# Incorporating Structural Motifs as Weak Priors in Molecular Learning

## Abstract

Deep learning models have shown themselves as a powerful solution for molecular property prediction, yet they are underutilized in real-world applications. These models, while powerful, lack chemical interpretability to link predicted properties to molecular motifs that govern them. Therefore, we introduce **C**hemically **I**nformed **L**anguage **T**ransformer (**CILT**) that utilizes hundreds of programmatically derived molecular motifs as a weak supervision prior. CILT leverages these motifs together with property descriptions to generate a chemically interpretable embedding space that clusters with respect to chemical motifs. This unified design enables CILT to quickly adapt to new motifs, properties, and perform classification, regression, and conditional generation. CILT showcases competitive performance while increasing the interpretability and requiring a 2-3 orders of magnitude fewer molecules.

## 1 Introduction

Chemists have long understood molecular behaviour through explicit linking of structural motifs — functional groups and molecular substructures — with observable properties. Machine learning models learn these relationships through data, often obscuring the underlying chemical relationships. The trade-off between the interpretability of hand-crafted features and the flexibility of machine learning remains unresolved.

Chemists represent these hand-crafted features as molecular fingerprints — vectors encoding the presence or count of functional groups, aromatic rings, and polar surface area. When crafted correctly, these fingerprints achieve high interpretability, sample efficiency, and outperform deep learning models (Praski et al., 2025; Boldini et al., 2024; Dekker et al., 2023). However, they are static; domain scientists maintain dozens of specialized fingerprints, each optimized for a specific property, making them inflexible and slow to adapt to new tasks. By design, they impose a hard inductive bias on the model and limit expressivity.

Deep learning methods learn molecular representations with little to no inductive bias, offering a layer of flexibility, unatainable to fingerprints. However, this comes at a cost of interpretability, making it hard to link underlying molecular motifs to predicted properties. This limits the applicability of deep learning models in real-world applications, where understanding the reasoning behind the prediction is as crucial as the prediction itself (Jiménez-Luna et al., 2020).

Here, we tackle this interpretability-flexibility trade-off by introducing a task-conditioned, motif-based, pre-training, creating an adaptive and interpretable molecular representation model. Our Chemically Informed Language Transformer (CILT) is pre-trained in a weakly supervised fashion on 300+ chemical motifs, programmatically derived from chemically important substructures and functional groups. These motifs are combined with natural text descriptors, enabling CILT to derive task-dependent embeddings. Furthermore, this design enables CILT to generate new molecules, predict properties, and adapt to any number of classification and regression tasks without changing the model's architecture or vocabulary, creating a unified, interpretable model for chemistry.

We demonstrate these capabilities through direct comparison with baseline models showcasing the interpretability with clear functional group clusters for CILTs embeddings and attention routes to relevant atoms. Zero-shot inference on novel tasks reaches 68% accuracy, and performance on

standard benchmarks remains competitive, demonstrating that interpretability and task-adaptivity do not sacrifice prediction quality.

Our main contributions are:

1. **Weakly supervised ask-adaptive pre-training framework**: We introduce a weak supervision approach on chemically important motifs paired with an open source library, ChemCaption, for molecular featurization.

2. **Chemical interpretability**: Functional group clustering, feature attribution, and task-conditioned attention routing demonstrate that learned representations align with chemical concepts and can be visualized to understand model reasoning per task.

3. **Semantic task reasoning capabilities**: Zero-shot inference (up to 68% on novel tasks) through interpretable task embeddings—capabilities categorically unavailable to task-agnostic pre-trained models that require fine-tuning on each new task.

4. **Data efficient competitive performance**: CILT outperforms other task-agnostic models on the MoleculeNet benchmark across multiple versions of the model, showing that structured weak supervision substitutes for scale without sacrificing prediction quality.

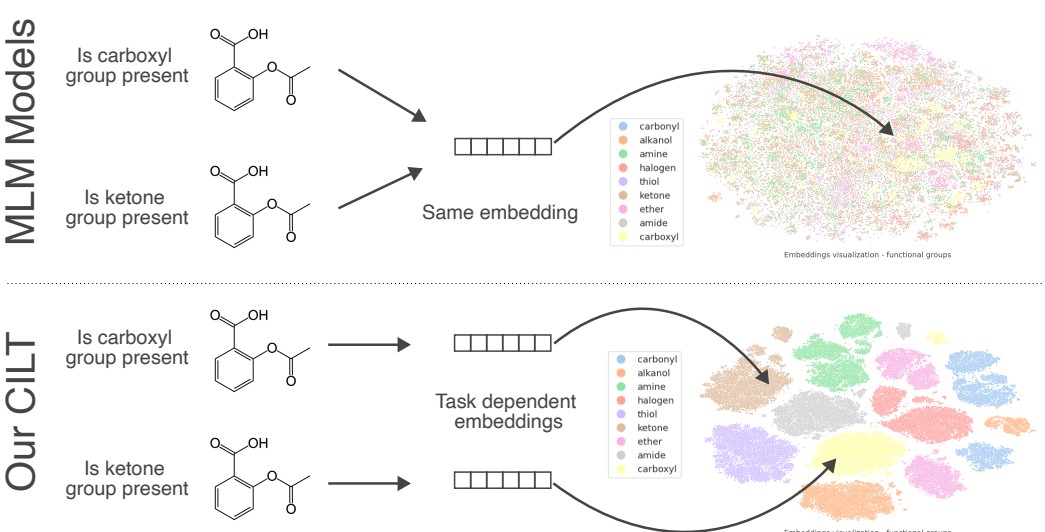

Figure 1: **Overview figure of different approaches.** Our CILT model can utilize the advantages of both deep learning and classical fingerprinting approaches.

## 2 BACKGROUND

**Group Contribution Methods** are a family of techniques for estimating molecular properties based on their substructural composition (Joback & Reid, 1987; Fredenslund et al., 1975). Molecules are decomposed into predefined structural groups, where each group has assigned empirically derived parameters that represent their contribution. These contributions are then combined, while accounting for the correction terms for group interactions, to form a property prediction. Chemists apply this method to this day to quickly and at scale estimate properties for mixture thermodynamics (Fredenslund et al., 1975), property estimation (Lydersen, 1955), drug discovery (Andrews et al., 1984), to name a few. Besides predictive power, thanks to hand-tuned features, predictions made with group-contribution approaches are very interpretable.

**Molecular Fingerprints** describe a molecule as a vector encoding the presence or count of predefined structural features. These fingerprints can then be used for fast similarity comparisons, forming the basis for structure-to-property predictive modeling. Machine learning models often offer

negligible gains compared to fingerprints while lacking interpretability and introducing additional computational overhead (Praski et al., 2025; Boldini et al., 2024).

# 3 RELATED WORK

**Molecular Representation Learning** Molecular property prediction has been addressed through diverse representation learning approaches. Sequence-based methods treat molecules as sequences, typically using the SMILES notation (Weininger, 1988) or other line representations such as SELF-IES (Krenn et al., 2022; 2020). Early work applied recurrent neural networks to SMILES (Segler et al., 2018; Mayr et al., 2018; Goh et al., 2017), while more recent approaches use transformer architectures with masked language modeling objectives (Ahmad et al., 2022; Chithrananda et al., 2020; Ross et al., 2022; Fabian et al., 2020; Honda et al., 2019; Irwin et al., 2022; Born & Manica, 2023). ChemBERTa (Chithrananda et al., 2020) adapts RoBERTa to molecular data, while MolFormer (Ross et al., 2022) scales to more than a billion molecules using linear attention mechanisms.

**Graph-based Approaches** One can represent molecules as molecular graphs with atoms as nodes and bonds as edges. Message-passing neural networks (Gilmer et al., 2017; Scarselli et al., 2008) form the foundation for many architectures. Self-supervised approaches include contrastive learning methods like MolCLR (Wang et al., 2022b), GraphCL (You et al., 2020), GraphMAE (Hou et al., 2022), and GROVER (Rong et al., 2020).

**Multi-Task and Auxiliary Supervision** Several approaches incorporate additional supervision signals during pretraining. MolBERT (Fabian et al., 2020) combines masked language modeling with auxiliary tasks such as descriptor prediction. ChemBERTa-2 (Ahmad et al., 2022) adds multi-task regression on physico-chemical properties. MoMu (Su et al., 2022) trains jointly on molecular graphs and natural language descriptions.

**Text-Molecule Joint Modeling** Recent works explore the joint modeling of natural language and molecular representations. MolT5 (Edwards et al., 2022) adapts T5 to perform both molecule-to-text and text-to-molecule generation tasks. Text2Mol (Edwards et al., 2021) learns cross-modal embeddings between molecular graphs and textual descriptions. MoleculeSTM (Liu et al., 2022) and CLAMP (Seidl et al., 2023) use contrastive learning between molecules and text. CLAMP learns CLIP-style contrastive alignments between molecules and text to improve downstream activity prediction from natural language assay descriptions. Instruction-following approaches include Galactica (Taylor et al., 2022), ether0 (Narayanan et al., 2025), and MolecularGPT (Liu et al., 2024).

**Task Conditioning and Prompting** In scientific domains, task conditioning appears in protein modeling (Ferruz et al., 2022; Liu et al., 2023), drug design (Bagal et al., 2021; Born & Manica, 2023) and optimization (Wu et al., 2024). However, most molecular models use fixed task identifiers or classification heads rather than natural language descriptions.

In summary, prior molecular pretraining has been largely optimized for token- or sequence-level objectives on SMILES, often requiring massive corpora before substructure knowledge emerges. We instead weakly supervise *on chemistry* via task-conditioned targets, derived via inexpensive calculations described in natural language, and we couple this with a dual-masking objective that ties text semantics to molecular structure. Empirically, this yields competitive accuracy with far fewer pretraining molecules, strong few-shot transfer and high interpretability; theoretically, task-similarity and motif-sparsity analyses explain when and why these gains appear.

# 4 CHEMICALLY INFORMED TASK CONDITIONING

## 4.1 PROBLEM SETUP

We pre-train a single 150M-parameter transformer on hundreds of molecular motifs expressed as natural language descriptors. Each task $t$ has a programmatic supervision function $g_t$ that extracts chemical properties from molecules: substructure indicators ("contains halogen group"), counts ("number of aromatic rings"), or simple properties ("molecular mass").

We unify the tasks and molecules by encoding them into text and jointly passing them throughout our network in the following form

$$\underbrace{d}_{\text{task description}} \quad [\text{SEP}] \quad \underbrace{y_t}_{\text{value tokens}} \quad [\text{SEP}] \quad \underbrace{x}_{\text{SMILES}}$$

This format enables conditional training, the model learns to predict masked SMILES tokens given properties and masked property values given SMILES, enabling a seamless switch between property prediction and generation as well as addition of new tasks.

## 4.2 Training Objective

We train with two alternating masked language modeling objectives. The SMILES objective (Equation (1)) teaches the model to generate molecules conditioned on task descriptions and target property values:

$$\mathcal{L}_{\text{SMILES}}(\theta) = \mathbb{E}_{t,x,M_x} \left[ - \sum_{i \in M_x} \log p_\theta(x_i \mid x_{\setminus i}, y_t, d_t) \right] \tag{1}$$

The property value objective (Equation (2)) teaches property prediction conditioned on molecular structure and task description:

$$\mathcal{L}_{\text{value}}(\theta) = \mathbb{E}_{t,x,M_y} \left[ - \sum_{j \in M_y} \log p_\theta(y_{t,j} \mid x, d_t) \right] \tag{2}$$

This bidirectional training creates a unified architecture for conditional generation, regression, and classification driven entirely by natural language prompts.

## 4.3 Theoretical Foundations

We provide theoretical justification for two key claims: why semantic similarity between task descriptions should predict transfer performance, and motif pretraining tasks should improve sample efficiency.

### 4.3.1 Task Similarity Controls Transfer

We first formalize the intuition that semantically similar task descriptions should enable better zero-shot transfer. We define this semantic similarity as the cosine similarity between task description embeddings: $s(d, d') = \langle e(d), e(d') \rangle$.

**Theorem 1** (Task-Semantic Adaptation Bound). *Under standard Lipschitz and bounded loss assumptions, the domain error $R$ on a target task $d'$ is bounded by:*

$$R_{d'}(h) \leq \underbrace{\sum_{t=1}^{T} \alpha_t R_{d_t}(h)}_{\text{weighted source risk}} + L \underbrace{\sum_{t=1}^{T} \alpha_t \|e(d') - e(d_t)\|}_{\text{task geometry term}} + \underbrace{\mathcal{O}(\sqrt{1/n})}_{\text{few-shot term}} \tag{3}$$

*for any convex combination of source tasks $\{\alpha_t\}$ and constant $L > 0$.*

*Where $R_{d_t}(h)$ represents the model's source domain error (on the pre-training tasks) while $R_{d'}(h)$ represents model's target domain error (on new tasks). The task embedding is represented as $e(d)$ and $\alpha_t$ is chosen via softmax over distance $\alpha_t = softmax(\|e(d') - e(d_t)\|)$.*

*Proof.* See section section A.1.

The *task geometry term* shows that transfer performance degrades with the distance between task embeddings. For unit-norm embeddings, $\|e(d') - e(d_t)\|^2 = 2(1 - \cos \angle(e(d'), e(d_t)))$, higher

cosine similarity implies better transfer when the weighted source domain errors are appropriately regularized. This provides theoretical backing for our empirical observation that zero-shot performance correlates with semantic similarity and indicates the number of shots needed to adapt to a re-phrased or related task (see Section 6.2).

### 4.3.2 MOTIF PRETRAINING IMPROVES SAMPLE EFFICIENCY

Next, we establish that when molecular properties depend on sparse combinations of motifs (e.g., functional groups) weak supervision on chemical motifs dramatically reduces sample complexity. This is a chemically informed prior based on the realization that chemists have achieved much success with so-called group contribution methods (Gani, 2019; Kühne et al., 1995; Constantinou & Gani, 1994; Fredenslund, 2012), where a property is predicted based on a linear or higher-order combination of group-specific factors (see Background section).

Suppose the pre-trained representations are *motif-aligned*, where motifs might correspond to functional group features, and suppose downstream molecular properties depend on sparse combinations of $k \ll p$ motifs, as suggested by the group contribution method. Under standard sparse regression assumptions:

**Theorem 2** (Motif Sample Complexity). *When molecular properties depend on $k$ motifs out of $p$ total features, explicit motif supervision reduces sample complexity from $\tilde{\mathcal{O}}(p/\varepsilon^2)$ to $\tilde{\mathcal{O}}(k \log p/\varepsilon^2)$ for achieving prediction error $\varepsilon$.*

*Proof.* See section section A.2.

## 5 METHODS

### 5.1 DATASET CONSTRUCTION

We construct our pretraining dataset by programmatically generating chemical task-property pairs from half a million diverse molecules from ChemPile-MLift (Mirza et al., 2025) using the Chem-Caption package, which interfaces with RDKit (Landrum, 2006). Our property set spans atom and bond counts, manually curated functional group indicators, ring system features, molecular descriptors, hydrogen bonding patterns, and substructure motifs. This yields over 300 distinct chemical properties per molecule.

Task descriptions are generated using templated natural language patterns. Task descriptions use templates like "`does the molecule contain ⟨PROPERTY_NAME⟩`" or "`what is the ⟨PROPERTY_NAME⟩ `", or "`number of ⟨PROPERTY_NAME⟩`". Property values are serialized as text tokens: binary values as "1"/"0", integers directly, and continuous values are first normalized and then quantized to four decimal places. This process generates approximately 150 million task-molecule pairs.

### 5.2 MODEL ARCHITECTURE AND TRAINING

We employ a 150M-parameter ModernBERT architecture (Warner et al., 2025) with a shared vocabulary combining SMILES tokens derived using a regular expression-based tokenizer (Schwaller et al., 2018), as well as natural language tokens, and numerical value tokens derived from the ModernBERT tokenizer. Input sequences follow the format `[task description] [SEP] [property value] [SEP] [SMILES]` with a maximum sequence length of 1024. Throughout all of the experiments no sequence has exceeded this limit.

Training alternates between the SMILES objective (Equation (1)) and the property prediction objective (Equation (2)) every 20 batch steps. The property prediction objective masks the entire property value and predicts it conditioned on the task description and SMILES sequence. The SMILES completion objective randomly masks 25% of the SMILES tokens and predicts them conditioned on the description of the task and the value of the property. Both objectives use cross-entropy loss with uniform task sampling across our property collection. We train the model for 3 epochs, for parameter breakdown see Section A.5.

## 5.3 Baselines

For comparison, we consider the following leading large chemical pretrained models: Mol-CLR (Wang et al., 2022a), ChemBERTa (Chithrananda et al., 2020), MolFormer (Ross et al., 2022), MolBert (Fabian et al., 2020), Grover (Rong et al., 2020), MolT5 (Edwards et al., 2022) and MoleculeSTM (Liu et al., 2022). We test all models on the MoleculeNet benchmark (Wu et al., 2018) and photoswitch dataset (Griffiths et al., 2022) (detailed description can be found in Section A.3.1 and Section A.3.2, respectively).

In the linear probe experiments, we train linear regression models for the regression tasks and logistic regression models for the classification tasks. For both, we utilize $L_1$ regularization (with optimal parameters see section A.2), additionally, for the logistic regression we employ the liblinear solver and balanced class weights. For all experiments, we use 4-fold cross-validation with scaffold splitting.

## 6 Experiments and Results

To demonstrate the effectiveness of our method, we evaluate CILT on multiple standard benchmarks in multiple systematic experiments: **a) embedding alignment** assessing the alignment of embeddings with chemically relevant features; **b) zero-/few-shot transfer** evaluating the performance of CILT on unseen tasks and the amount of data needed for adaptation to these tasks; **c) linear probes** comparing embeddings across different models to evaluate innate learned molecular representations; **d) ablations** for targeted assessment of our training methodology.

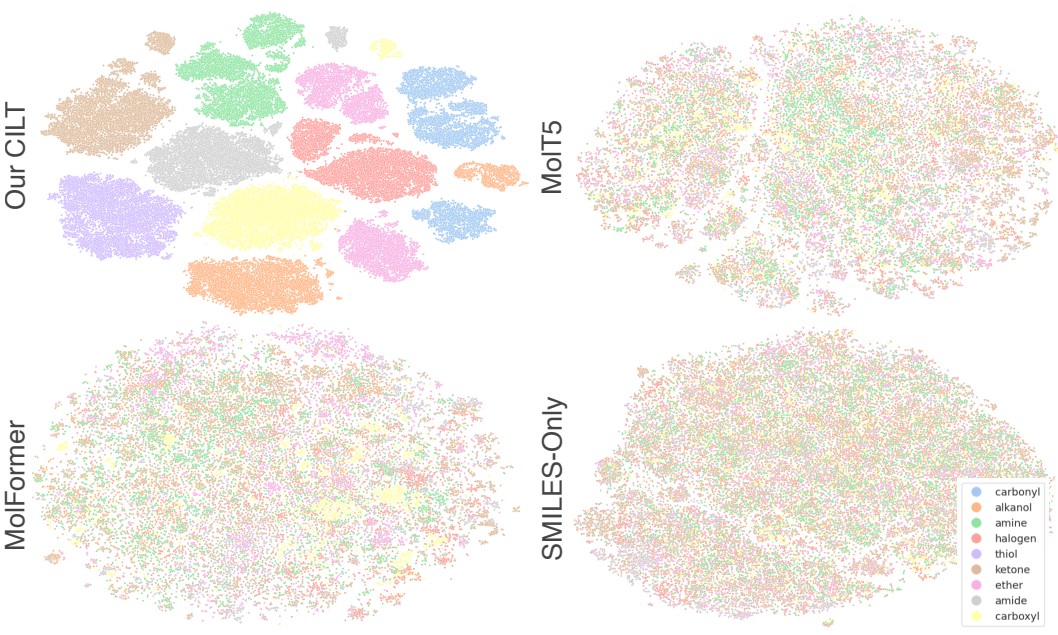

Figure 2: **Visualization of learned embeddings represented via t-SNE.** Representations are extracted from the hold-out test set (scaffold-split) used for the pre-training of CILT. The models used for comparisons are MolFormer (Ross et al., 2022), MolT5 (Edwards et al., 2022) and a SMILES-only pre-trained version of CILT.

### 6.1 Representations Alignment with Functional Groups

**Experiment** To show the benefit of pre-training with molecular fingerprints as a weak prior we compare the embeddings across CILT, SMILES only trained version of CILT, MolFormer (Ross et al., 2022) and MolT5 (Edwards et al., 2022) Figure 2.

**Results** Figure 2 show that CILT's embeddings cluster align with the presence and absence of functional groups, while the classical MLM approaches and multimodal modeling are not able to make this distinction. This indicates that the model can capture relevant chemical features that are known to chemists and therefore offers a higher level of interpretability. This gives additional support to our assumption that we induce motif-aligned coordinates in our representation (see Theorem 2). For a full breakdown of per-functional group embeddings see Section A.7.

Additionally, we also find attention patterns to show chemically meaningful behaviors (Section A.6). Chemically relevant atoms have higher attention scores, and attention patterns link the task to the property and then to relevant atoms.

## 6.2 Zero-Shot Transfer

**Experiment** Theorem 1 predicts that semantically similar task descriptions should enable better zero-shot transfer. To evaluate this, we conducted an experiment on a subset of functional group presence tasks. We rephrase the original template 20 times (see Section A.4) and measure the cosine similarity between the new and original task description. We then group the tasks by cosine similarity and evaluate the model on them. First, we measure the zero-shot performance, and then we gradually increase the number of fine-tuning data points until all of the tasks converge. Baseline models do not have the ability to adapt to new task descriptions, therefore we report only their zero-shot performance.

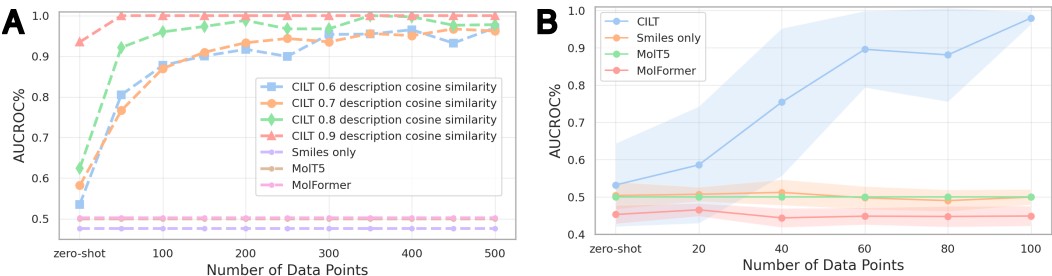

Figure 3: **Adaptation to new tasks. A** Adaptation to the new task description for the already seen task based on the cosine similarity to the original task description. Baseline models do not have the ability to adapt to new task descriptions, their zero-shot performance is reported. **B** Required number of data points to adapt to the unseen tasks across 12 methylations.

**Results** Figure 3 **A** shows that across all of the datasets the cosine similarity is correlated both in the zero-shot performance and the adaptation setting. We see that the datasets with higher cosine similarity between the new task description and the original task description from pretraining adapt with fewer data points. This gives support to our assumptions that semantically similar task descriptions should enable better zero-shot transfer. Furthermore, none of the baseline models show zero-shot performance better than random, indicating that the models are not capable of distinguishing between different functional groups.

## 6.3 Few-shot transfer

**Experiment** Theorem 2 predicts that motif-alignment leads to more data efficient learning. We test this by altering the original task. We perform methylations (replacing one H with $CH_3$) on the substructures that CILT has been trained to understand. We gathered 15 of these new tasks to evaluate our model. After evaluating the zero-shot, we gradually increase the number of training points by 20 (10 positive and 10 negative samples) until our models converge. For the baseline models, we freeze the backbone and replace the last layer with the prediction head and fine-tune the model as a binary classifier.

**Results** Figure 3 **B** shows that across all of the methylations, CILT can fine-tune with less than 100 samples and even perform zero-shot inference in some settings. This gives support to our

assumptions that motif alignment leads to more data-efficient learning. Furthermore, none of the baseline models show performance significantly better than random at any step of the fine-tuning process.

## 6.4 TRANSFERABILITY OF THE EMBEDDINGS

**Experiment** We assess the robustness and transferability of the embeddings of CILT and other baseline encoders using linear probing (Alain & Bengio, 2016). We report the %AUCROC for classification tasks and MAE for regression tasks along with the standard deviations.

Table 1: **Embedding quality estimated using linear probes.** Logistic regression and linear regression trained on embeddings over 4-fold cross-validation scaffold split. For classification we report %AUCROC ($\uparrow$) and for regression MAE ($\downarrow$). The best results in each column are bolded and the second best are underlined. CILT is the best model for classification tasks.

| Classification (%AUCROC $\uparrow$) | | | | | | | | | |
|---|---|---|---|---|---|---|---|---|---|
| Model | BACE | BBBP | ClinTox | HIV | SIDER | Tox21 | ToxCast | MUV | Avg. |
| MolCLR | $73.4 \pm 3.6$ | $82.42 \pm 2.1$ | $70.5 \pm 3.7$ | $71.2 \pm 0.9$ | $58.9 \pm 4.8$ | $69.7 \pm 7.6$ | $62.5 \pm 10.1$ | $70.54 \pm 13.9$ | 69.9 |
| ChemBERTa | $80.0 \pm 3.6$ | $88.0 \pm 2.2$ | $97.2 \pm 1.5$ | $73.9 \pm 1.9$ | $\underline{54.1 \pm 6.0}$ | $67.8 \pm 6.8$ | $64.0 \pm 10.5$ | $72.8 \pm 11.1$ | 74.7 |
| MolFormer | $74.3 \pm 2.1$ | $89.8 \pm 1.0$ | $97.2 \pm 1.5$ | $73.9 \pm 0.9$ | $55.8 \pm 5.1$ | $68.0 \pm 6.2$ | $65.3 \pm 10.2$ | $71.9 \pm 15.7$ | 74.5 |
| Grover | $\mathbf{84.2 \pm 3.8}$ | $84.1 \pm 0.8$ | $82.8 \pm 3.1$ | $\mathbf{78.5 \pm 2.3}$ | $56.7 \pm 6.6$ | $71.3 \pm 6.6$ | $67.0 \pm 10.7$ | $73.8 \pm 12.6$ | 75.0 |
| MolBERT | $81.0 \pm 4.2$ | $82.9 \pm 2.2$ | $77.9 \pm 6.3$ | $75.4 \pm 2.2$ | $56.9 \pm 4.6$ | $70.4 \pm 6.9$ | $63.9 \pm 10.4$ | $\mathbf{76.2 \pm 12.8}$ | 73.1 |
| MolT5 | $81.9 \pm 3.5$ | $94.3 \pm 1.6$ | $97.4 \pm 2.7$ | $75.8 \pm 1.6$ | $\mathbf{60.3 \pm 7.8}$ | $\mathbf{74.0 \pm 6.7}$ | $\mathbf{69.9 \pm 10.4}$ | $74.0 \pm 13.9$ | **78.4** |
| MoleculeSTM | $73.7 \pm 4.2$ | $\underline{87.6 \pm 1.9}$ | $\underline{98.0 \pm 0.6}$ | $71.1 \pm 1.0$ | $56.3 \pm 5.2$ | $69.6 \pm 6.2$ | $64.2 \pm 10.7$ | $67.4 \pm 11.8$ | 73.5 |
| CILT(500k) | $80.4 \pm 1.2$ | $92.5 \pm 1.2$ | $\underline{97.7 \pm 1.5}$ | $73.9 \pm 1.5$ | $55.2 \pm 6.3$ | $66.3 \pm 6.9$ | $64.4 \pm 10.3$ | $71.9 \pm 13.7$ | 75.3 |
| CILT(250k) | $81.3 \pm 2.5$ | $\mathbf{94.5 \pm 1.3}$ | $\mathbf{98.3 \pm 0.1}$ | $75.6 \pm 0.7$ | $58.5 \pm 6.8$ | $\underline{72.5 \pm 6.0}$ | $\underline{68.0 \pm 11.2}$ | $\underline{75.2 \pm 12.3}$ | 78.0 |

| Regression (MAE $\downarrow$) | | | | | | | | | |
|---|---|---|---|---|---|---|---|---|---|
| Model | Lipo | ESOL | FreeSolv | CAM | PBE0 | $En - \pi*$ | $E\pi - \pi*$ | $Zn - \pi*$ | Rank |
| MolCLR | $1.00 \pm 0.04$ | $1.03 \pm 0.09$ | $1.16 \pm 0.34$ | $36.7 \pm 21.3$ | $\mathbf{37.5 \pm 7.9}$ | $25.8 \pm 12.9$ | $50.5 \pm 7.7$ | $\underline{13.8 \pm 5.3}$ | 3.0 |
| ChemBERTa | $\underline{0.81 \pm 0.30}$ | $0.82 \pm 0.73$ | $0.86 \pm 0.27$ | $34.2 \pm 21.1$ | $43.4 \pm 16.1$ | $26.7 \pm 12.3$ | $\mathbf{47.3 \pm 10.6}$ | $\underline{13.8 \pm 5.3}$ | $\underline{2.0}$ |
| MolFormer | $\underline{0.81 \pm 0.04}$ | $0.83 \pm 0.73$ | $0.88 \pm 0.23$ | $43.1 \pm 12.3$ | $55.2 \pm 14.2$ | $26.9 \pm 12.3$ | $50.9 \pm 9.1$ | $\underline{13.8 \pm 5.3}$ | 3.8 |
| Grover | $\underline{0.81 \pm 0.03}$ | $0.82 \pm 0.73$ | $\mathbf{0.85 \pm 0.27}$ | $39.8 \pm 23.3$ | $44.6 \pm 18.0$ | $23.5 \pm 8.7$ | $67.5 \pm 11.1$ | $16.5 \pm 5.2$ | 3.0 |
| MolBERT | $1.00 \pm 0.04$ | $\underline{1.03 \pm 0.08}$ | $1.64 \pm 0.34$ | $47.0 \pm 25.8$ | $\underline{41.5 \pm 21.8}$ | $31.0 \pm 11.3$ | $58.6 \pm 10.3$ | $16.6 \pm 5.0$ | 5.4 |
| MolT5 | $0.81 \pm 0.03$ | $0.82 \pm 0.73$ | $0.86 \pm 0.27$ | $\mathbf{33.3 \pm 17.7}$ | $\underline{43.7 \pm 15.2}$ | $24.7 \pm 13.5$ | $47.4 \pm 12.1$ | $\underline{13.8 \pm 5.3}$ | **1.9** |
| MoleculeSTM | $\underline{0.81 \pm 0.03}$ | $\underline{0.82 \pm 0.73}$ | $\underline{0.86 \pm 0.27}$ | $44.1 \pm 15.3$ | $55.0 \pm 12.1$ | $27.3 \pm 12.0$ | $50.6 \pm 7.8$ | $\underline{13.8 \pm 5.3}$ | 3.6 |
| CILT(500k) | $\mathbf{0.80 \pm 0.02}$ | $0.88 \pm 0.18$ | $0.91 \pm 0.30$ | $46.9 \pm 15.5$ | $58.5 \pm 7.6$ | $27.5 \pm 12.0$ | $51.3 \pm 7.3$ | $13.9 \pm 5.2$ | 4.9 |
| CILT(250k) | $\underline{0.81 \pm 0.02}$ | $0.90 \pm 0.18$ | $0.91 \pm 0.30$ | $49.1 \pm 19.1$ | $65.8 \pm 7.0$ | $27.5 \pm 12.0$ | $51.3 \pm 7.3$ | $13.9 \pm 5.2$ | 5.1 |

**Results** Table 4 shows that CILT demonstrates competitive performance across all the datasets. In the classification setting, it achieves the 2 best and 3 second best scores, while in the regression setting, it shows the second-to-last performance. We theorise that the lackluster performance on regression tasks comes from pre-training being dominated by classification tasks.

## 6.5 ABLATIONS

**Experiment** To isolate the effect of task conditioning, we train a control model using identical architecture and hyperparameters but with standard masked language modeling on SMILES sequences only, without task descriptions or property values. This control methodology represents conventional molecular pretraining approaches like ChemBERTa and MolFormer.

We evaluate both the task-conditioned model and the SMILES-only baseline on the same downstream benchmarks using identical fine-tuning protocols.

**Results** Table 2 shows that task-conditioned pretraining outperforms SMILES-only pretraining on 15 out of 16 tasks across two benchmark datasets. This confirms that our chemically meaningful pretraining tasks provide measurable benefits over standard molecular language modeling.

## 7 DISCUSSION

**Parameter–Performance Frontier** In Figure 4, we plot the average classification performances from the linear probe experiments (Section 6.4) and compare them against the log number of

Table 2: **Ablation results.** Logistic regression and linear regression trained on embeddings over a 4-fold cross-validation scaffold split. For classification we report %AUCROC (↑) and for regression MAE (↓). The best results are bolded. We find that CILT outperforms the SMILES-only model on both classification and regression tasks.

| | Classification (%AUCROC ↑) | | | | | | | | |
|---|---|---|---|---|---|---|---|---|---|
| Model | BACE | BBBP | ClinTox | HIV | SIDER | tox21 | ToxCast | MUV | Mean |
| SmilesOnly | $74.7 \pm 2.3$ | $90.5 \pm 1.1$ | $97.3 \pm 2.0$ | $70.1 \pm 1.2$ | $55.2 \pm 6.1$ | $65.7 \pm 6.6$ | $63.4 \pm 10.1$ | $68.7 \pm 13.7$ | 73.2 |
| CILT(500k) | $80.4 \pm 1.2$ | $92.5 \pm 1.2$ | $97.7 \pm 1.5$ | $73.9 \pm 1.5$ | $55.2 \pm 6.3$ | $66.3 \pm 6.9$ | $64.4 \pm 10.3$ | $71.9 \pm 13.7$ | 75.3 |
| CILT(250k) | $\mathbf{81.3 \pm 2.5}$ | $\mathbf{94.5 \pm 1.3}$ | $\mathbf{98.3 \pm 0.1}$ | $\mathbf{75.6 \pm 0.7}$ | $\mathbf{58.5 \pm 6.8}$ | $\mathbf{72.5 \pm 6.0}$ | $\mathbf{68.0 \pm 11.2}$ | $\mathbf{75.2 \pm 12.3}$ | 78.0 |

| | Regression (MAE ↓) | | | | | | | | |
|---|---|---|---|---|---|---|---|---|---|
| Model | Lipo | FreeSolv | ESOL | CAM | PBE0 | $En - \pi*$ | $E\pi - \pi*$ | $Zn - \pi*$ | Rank |
| SmilesOnly | $0.81 \pm 0.03$ | $0.89 \pm 0.07$ | $\underline{0.91 \pm 0.18}$ | $49.2 \pm 16.9$ | $77.4 \pm 15.3$ | $30.0 \pm 11.9$ | $62.8 \pm 6.9$ | $17.1 \pm 4.8$ | 2.6 |
| CILT(500k) | $\mathbf{0.80 \pm 0.02}$ | $\mathbf{0.88 \pm 0.18}$ | $\underline{0.91 \pm 0.30}$ | $\mathbf{46.9 \pm 15.5}$ | $\mathbf{58.5 \pm 7.6}$ | $\underline{27.5 \pm 12.0}$ | $\underline{51.3 \pm 7.3}$ | $\underline{13.9 \pm 5.2}$ | **1.1** |
| CILT(250k) | $0.81 \pm 0.02$ | $0.90 \pm 0.18$ | $\underline{0.91 \pm 0.30}$ | $49.1 \pm 19.1$ | $65.8 \pm 7.0$ | $\underline{27.5 \pm 12.0}$ | $\underline{51.3 \pm 7.3}$ | $\underline{13.9 \pm 5.2}$ | 1.6 |

molecules used in pre-training. Our model CILT shows competitive performance across multiple versions while only requiring a fraction of molecules. This challenges the assumption that sequence-based molecular foundation models need to be trained on a huge number of molecules to work well.

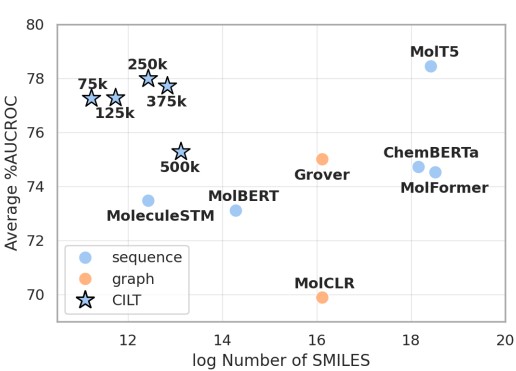

Figure 4: **Log number of pretraining molecules vs. downstream performance.** We show the number of molecules used in pretraining of baseline models and CILT vs. the average classification performance of linear probes on MoleculeNet. CILT shows the best tradeoff between dataset size and performance.

**Meaningful Representations Through Soft Inductive Biases.** Our approach succeeds by implementing soft inductive biases—preferences for certain solutions without hard constraints (Wilson, 2025). Rather than restricting the model architecture, we guide learning through natural language task conditioning. This creates representations that cluster by chemically important features without explicit supervision, while attention mechanisms focus on chemically relevant atoms when processing task descriptions. Our theoretical analysis shows that semantic similarity between task descriptions directly predicts transfer performance (Theorem 2), while Theorem 1 formalizes how motif-based supervision reduces sample complexity from $\mathcal{O}(p)$ to $\mathcal{O}(k \lg p)$. The model learns chemical intuition not as an emergent property by scaling data, but as an explicit objective encoded through structured tasks.

**Task Conditioning as Architectural Innovation** The natural language conditioning framework offers practical advantages beyond efficiency. Unlike approaches that require architectural changes for new properties and downstream applications, our text-based task descriptions enable immediate extensibility. New chemical tasks can be incorporated without re-training by simply providing appropriate natural language descriptions, making the system immediately adaptable to new chemical properties.

**Future Directions** The current CILT model is pre-trained on a naive selection of motifs and task descriptions; therefore, the next future step would be to improve the selection of pre-training motifs and rephrase the task descriptions (Maini et al., 2024; Pieler et al., 2024). The semantic similarity results also suggest principled curriculum learning possibilities.

## 8 CONCLUSIONS

Foundation models (White, 2023; Ramos et al., 2025; Alampara et al., 2025) for scientific domains commonly follow the standard approach following the NLP blueprint: scale data and parameters until patterns emerge (Frey et al., 2023). But scientific domains differ fundamentally from language. Chemical datasets are small, diverse, and experimental data is expensive. But scientific domains possess structured theoretical knowledge that language modeling lacks. In chemistry, for instance, this has been encoded over decades via QSPR relationships and group contribution theory. Rather than rediscovering them from data, we can use them as a weak supervision signal.

We demonstrate that chemically-informed pretraining achieves competitive performance with orders of magnitude less data. By encoding chemical priors as soft inductive biases through natural language task conditioning, CILT learns interpretable representations that respect chemical structure while enabling rapid adaptation to new tasks.

Our approach of pre-training on a broad basis of weakly supervised tasks in multiple masking objectives might be a recipe for other domains where there is little data, but one can generate tasks with some weak-supervision-like techniques.

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

# A APPENDIX

## A.1 PROOF OF THEOREM 1: TASK-SEMANTIC ADAPTATION BOUND

We prove that the domain error on a target task can be bounded in terms of source task performance plus a term that depends on the semantic similarity between task descriptions. The key insight is to use optimal transport theory to relate distributional differences to task embedding distances.

Let $P_d$ denote the joint distribution of motif-task pairs $(\phi_\theta(X, d), g_d(X))$ for task $d$, where $X \sim \mathcal{D}$. The domain error under task $d$ is $R_d(h) = \mathbb{E}_{(Z,Y) \sim P_d}[\ell(h(Z), Y)]$.

**Step 1: Kantorovich-Rubinstein bound.** We want to bound the difference in domain error between the target task $d'$ and a weighted combination of source tasks $d$. Since the loss function $\ell$ is $L_f$-Lipschitz by assumption, we can apply the Kantorovich-Rubinstein duality, which provides a connection between differences in expectations and Wasserstein distances (Villani, 2008):

$$\left| R_{d'}(h) - \sum_{t=1}^{T} \alpha_t R_{d_t}(h) \right| = \left| \mathbb{E}_{P_{d'}}[\ell(h(Z), Y)] - \mathbb{E}_{\sum_t \alpha_t P_{d_t}}[\ell(h(Z), Y)] \right| \leq L_f W_1\left(P_{d'}, \sum_{t=1}^{T} \alpha_t P_{d_t}\right).$$

This converts the problem from bounding differences in domain errors to bounding Wasserstein distances between distributions (which is a geometric problem about the learned representations).

**Step 2: Pushforward representation.** The joint distributions $P_d$ arise from our specific model architecture. We can represent them as pushforwards of simpler distributions through our learned mapping.

Define the map $\Psi : (x, u) \mapsto (\phi_\theta(x, d(u)), g_{d(u)}(x))$ that transforms molecules and task embeddings into motifs and tasks. This map encapsulates both our learned representation function and the ground truth property computation.

Since each task $d$ corresponds to a fixed task embedding $e(d)$, we can write:

$$P_d = \Psi_\#(\mathcal{D} \otimes \delta_{e(d)}),$$

where $\Psi_\#$ denotes the pushforward measure. This means the distribution $P_d$ is obtained by taking the product of the molecular distribution $\mathcal{D}$ with a point mass at the task embedding $e(d)$, then applying the transformation $\Psi$.

For the weighted combination of source distributions:

$$\sum_{t=1}^{T} \alpha_t P_{d_t} = \Psi_\# \left( \mathcal{D} \otimes \sum_{t=1}^{T} \alpha_t \delta_{e(d_t)} \right).$$

**Step 3: Wasserstein contraction.** Now we can use the property that the Wasserstein distance contracts under Lipschitz maps. By assumption, the map $\Psi$ is $L_\Psi$-Lipschitz in the task embedding component. This means that if two task embeddings are close, the resulting motif-task distributions will also be close.

The contraction property gives us:

$$W_1 \left( P_{d'}, \sum_{t=1}^{T} \alpha_t P_{d_t} \right) \leq L_\Psi W_1 \left( \mathcal{D} \otimes \delta_{e(d')}, \mathcal{D} \otimes \sum_{t=1}^{T} \alpha_t \delta_{e(d_t)} \right)$$

Since the molecular distribution $\mathcal{D}$ is the same in both cases, the Wasserstein distance only depends on the task embedding component:

$$W_1 \left( \delta_{e(d')}, \sum_{t=1}^{T} \alpha_t \delta_{e(d_t)} \right) = \sum_{t=1}^{T} \alpha_t \|e(d') - e(d_t)\|$$

The distributional distance between tasks thus reduces to the geometric distance between their embeddings. This justifies why semantic similarity should predict transfer performance.

**Step 4: Finite-sample bound.** Finally, we need to account for the fact that we only have finite samples from the target task. The standard approach uses Rademacher complexity to bound the gap between empirical and population risk. For bounded loss functions and hypothesis class $\mathcal{H}$, concentration inequalities give (Mohri et al., 2018):

$$R_{d'}(h) \leq \hat{R}_{d'}(h) + 2\mathfrak{R}_n(\mathcal{H}) + 3\sqrt{\frac{\ln(2/\delta)}{2n}}$$

with probability at least $1 - \delta$, where $\hat{R}_{d'}(h)$ is the empirical domain error on the target task.

Combining all steps and minimizing the empirical term over $h \in \mathcal{H}$ yields the bound in Theorem 1. The interpretation is that target task performance is bounded by a weighted combination of source performance, plus a penalty term proportional to the distance between task embeddings, plus a finite-sample correction. $\qquad \square$

A.2    PROOF OF THEOREM 2: MOTIF SAMPLE COMPLEXITY

We analyze when explicit motif supervision can reduce sample complexity compared to standard dense regression. The key insight is that chemical properties often depend on sparse combinations of motifs, making this a sparse regression problem where $k \ll p$ motifs matter.

**Setup and intuition.** Consider a pretrained encoder that produces representations $\psi_\theta(x) \in \mathbb{R}^p$ that are *motif-aligned*—meaning different coordinates respond to different motifs. If downstream molecular properties depend on only $k$ out of $p$ possible motifs, then the optimal linear head $w^\star$ should be $k$-sparse.

We analyze the LASSO estimator (Tibshirani, 1996), which is designed to recover sparse solutions:

$$\hat{w} \in \arg\min_{w \in \mathbb{R}^p} \frac{1}{2n}\|y - \Psi w\|_2^2 + \lambda\|w\|_1$$

where $\Psi \in \mathbb{R}^{n \times p}$ stacks rows $\psi_\theta(x_i)^\top$ and $y_i = w^{\star\top}\psi_\theta(x_i) + \xi_i$.

**Step 1: Basic inequality.** The proof follows the standard template for LASSO analysis. By optimality of $\hat{w}$, it achieves lower objective value than the true parameter $w^\star$:

$$\frac{1}{2n}\|y - \Psi\hat{w}\|_2^2 + \lambda\|\hat{w}\|_1 \le \frac{1}{2n}\|y - \Psi w^\star\|_2^2 + \lambda\|w^\star\|_1$$

Since $y = \Psi w^\star + \xi$ where $\xi$ is noise, we can expand and simplify to get:

$$\frac{1}{2n}\|\Psi\Delta\|_2^2 \le \frac{1}{n}\xi^\top\Psi\Delta + \lambda(\|w^\star\|_1 - \|\hat{w}\|_1),$$

where $\Delta = \hat{w} - w^\star$ is the estimation error.

The left side is the prediction error, while the right side has a stochastic term and a regularization term.

**Step 2: Controlling the stochastic term.** The term $\frac{1}{n}\xi^\top\Psi\Delta$ involves the noise and is the main source of randomness. We can bound it using the dual norm relationship:

$$\frac{1}{n}\xi^\top\Psi\Delta = \left\langle \frac{1}{n}\Psi^\top\xi, \Delta \right\rangle \le \left\|\frac{1}{n}\Psi^\top\xi\right\|_\infty \|\Delta\|_1$$

Since the noise $\xi$ is sub-Gaussian, concentration inequalities tell us that with high probability:

$$\left\|\frac{1}{n}\Psi^\top\xi\right\|_\infty \le C\sigma\sqrt{\frac{\log p}{n}}$$

We choose the regularization parameter $\lambda$ to be twice this bound, so that:

$$\frac{1}{n}\xi^\top\Psi\Delta \le \frac{\lambda}{2}\|\Delta\|_1$$

This is a standard technique in high-dimensional statistics: choose $\lambda$ large enough to dominate the stochastic fluctuations.

**Step 3: Decomposability and cone constraint.** Now we analyze the regularization term $\|w^\star\|_1 - \|\hat{w}\|_1$. Since $w^\star$ is $k$-sparse with support $S = \text{supp}(w^\star)$, we can decompose:

$$\|w^\star\|_1 - \|\hat{w}\|_1 = \|w_S^\star\|_1 - \|\hat{w}_S\|_1 - \|\hat{w}_{S^c}\|_1$$

Using the reverse triangle inequality $\|a\|_1 - \|a + b\|_1 \le \|b\|_1$:

$$\|w_S^\star\|_1 - \|\hat{w}_S\|_1 = \|w_S^\star\|_1 - \|w_S^\star + \Delta_S\|_1 \le \|\Delta_S\|_1$$

Therefore: $\|w^\star\|_1 - \|\hat{w}\|_1 \le \|\Delta_S\|_1 - \|\Delta_{S^c}\|_1$.

Combining with the previous steps gives:

$$\frac{1}{2n}\|\Psi\Delta\|_2^2 \le \frac{\lambda}{2}\|\Delta\|_1 + \lambda(\|\Delta_S\|_1 - \|\Delta_{S^c}\|_1) = \frac{3\lambda}{2}\|\Delta_S\|_1 - \frac{\lambda}{2}\|\Delta_{S^c}\|_1$$

Rearranging: $\|\Delta_{S^c}\|_1 \le 3\|\Delta_S\|_1$ (cone constraint).

**Step 4: Restricted eigenvalue and final bound.** The cone constraint (Hastie et al., 2015) allows us to control the estimation error using the restricted eigenvalue (RE) condition (Raskutti et al., 2010). This condition requires that the design matrix $\Psi$ has good properties when restricted to sparse vectors:

$$\frac{1}{n}\|\Psi\Delta\|_2^2 \geq \kappa\|\Delta\|_2^2$$

for all $\Delta$ satisfying the cone constraint.

The RE condition is natural for motif-aligned representations: it says that different motifs produce sufficiently different representation patterns that they can be distinguished statistically.

Using the Cauchy-Schwarz inequality $\|\Delta_S\|_1 \leq \sqrt{k}\|\Delta\|_2$ and combining with our earlier bound:

$$\frac{\kappa}{2}\|\Delta\|_2^2 \leq \frac{1}{2n}\|\Psi\Delta\|_2^2 \leq \frac{3\lambda}{2}\sqrt{k}\|\Delta\|_2$$

Solving: $\|\Delta\|_2 \leq \frac{3\sqrt{k}}{\kappa}\lambda$.

For the prediction error:

$$\frac{1}{n}\|\Psi(\hat{w} - w^\star)\|_2^2 \leq \frac{9k}{\kappa}\lambda^2$$

**Sample complexity conclusion.** With $\lambda = C\sigma\sqrt{\frac{\log p}{n}}$, achieving prediction error at most $\varepsilon^2$ requires:

$$\frac{9k}{\kappa} \cdot C^2\sigma^2\frac{\log p}{n} \leq \varepsilon^2$$

Solving for $n$:

$$n \geq \frac{9C^2\sigma^2 k \log p}{\kappa\varepsilon^2} = \tilde{\mathcal{O}}\left(\frac{\sigma^2}{\kappa} \cdot \frac{k \log p}{\varepsilon^2}\right).$$

This improves upon the standard dense regression bound of $\tilde{\mathcal{O}}(p/\varepsilon^2)$ by a factor of $p/(k \log p)$. When motifs are sparse ($k \ll p$), this represents an exponential improvement in sample complexity. $\square$

## A.3 DATA

We provide a short overview of the dataset used in this study.

### A.3.1 MOLECULENET

We use MoleculeNet Wu et al. (2018) as one of our benchmarks. All of the benchmarks are used with scaffold splitting. The benchmark contains the following datasets:

**BACE** BACE contains approximately 1.5k molecules and their bioactivity measurement for inhibition of human $\beta$-secretase 1 (BACE-1). The bioactivity values are an aggregate of scientific literature and not from a single bioassay.

**BBBP** The blood-brain barrier penetration dataset contains approximately 2k molecules, and its activity is determined by whether it is able to pass the highly selective membrane and enter the brain fluid.

**ClinTox** The clinical toxicity (ClinTox) contains two bioactivity prediction tasks: (1) FDA approval and (2) failure of clinical trials. The dataset contains approximately 58k molecules.

**HIV** The HIV dataset contains approximately 40k of molecules and measures the evidence of anti-HIV activity.

**SIDER**    The side effect resources (SIDER) dataset contains approximately 1.4k molecules spanning 27 assays measuring the side effects of drugs.

**Tox21**    The Tox21 dataset measures the drug-related effects spanning 12 different prediction tasks with over 7.8k molecules.

**ToxCast**    The ToxCast dataset provides 617 classification tasks based on in vitro drug screening. The dataset contains 8.5 molecules.

**MUV**    The maximum unbiased validation (MUV) dataset spans 17 tasks designed to identify active compounds. The dataset contains approximately 93k molecules.

**Lipo**    The lipophilicity dataset contains hydrophobicity measurements of 4.2k molecules.

**ESOL**    The Delaney Solubility Dataset contains water solubility measurements for over 1.1k of molecules.

**FreeSolv**    The Freesolv dataset contains the measurements for hydration free energy for small molecules and contains 624 molecules.

### A.3.2    PHOTOSWITCH

For additional regression tasks, we use the photoswitch dataset (Griffiths et al., 2022), where we use the datasets that contain more than 100 molecules, and we again scaffold-split the datasets.

**CAM**    The CAM-B3LYP benchmark contains 117 molecules and computed electronic transition wavelengths in nm.

**PBE0**    The PBE0 dataset contains 114 molecules and computed electronic transition wavelengths.

$E$ **and** $Z$ **isomer**    These datasets contain the wavelengths of transitions between different electronic states ($n$, $\pi$, $\pi*$) that have been observed for the different isomers.

### A.4    TEMPLATE REPHRASES

List of rephrased templates for functional groups used in Section 6.2. The $\langle$GROUP $\rangle$parameters are replaced with the name of the functional group:

- "is the $\langle$GROUP$\rangle$ group present"
- "does it have a $\langle$GROUP$\rangle$ group"
- "is there a $\langle$GROUP$\rangle$ group in it"
- "does this structure include a $\langle$GROUP$\rangle$ group"
- "is a $\langle$GROUP$\rangle$ group part of the molecule"
- "does the compound contain a $\langle$GROUP$\rangle$ group"
- "can a $\langle$GROUP$\rangle$ group be found here"
- "is the $\langle$GROUP $\rangle$functional group present"
- "does the molecule feature a $\langle$GROUP$\rangle$ group"
- "is there evidence of a $\langle$GROUP $\rangle$functional group"
- "does this molecule exhibit a $\langle$GROUP$\rangle$ group"
- "is a $\langle$GROUP $\rangle$functional group detectable"
- "does the structure show the presence of $\langle$GROUP $\rangle$"
- "can a $\langle$GROUP$\rangle$ group be identified here"
- "is $\langle$GROUP $\rangle$part of the chemical composition"

- "does the sample possess a ⟨GROUP⟩ group"
- "is there a ⟨GROUP ⟩moiety in this compound"
- "does this substance carry a ⟨GROUP⟩ group"
- "can the molecule be classified as containing a ⟨GROUP⟩ group"
- "is the ⟨GROUP ⟩function observed in this case"

## A.5 TRAINING PARAMETERS

Table 3: **Training hyperparameters.** Hyperparameter setting used to train our model.

| Hyperparameter | Value |
|---|---|
| Batch size | 76 |
| GPUs | 6 x NVIDIA H100 |
| GPUh | 252h |
| Alternating loss steps | 20 |
| Precision | float16 |
| Hidden size | 768 |
| Maximum of positional embeddings | 1024 |
| Number of hidden layers | 22 |
| Learning rate | 0.01 |
| Warmup steps | 10000 |
| Optimizer | AdaFactor (Shazeer & Stern, 2018) |

### A.6 ATTENTION

In Section A.6, we show an example of how attention maps the property value token to the description and the relevant atoms, in this case, that is Fluorine (F). Additionally, we show that the atom itself attends to a phrase "contains halogen" as well as the property value.

In Section A.6, we show the average attention per SMILES token across all attention heads for the second-to-last layer. The results are averaged over 5000 molecules that contain a halogen group, where we fix the task description as shown in Section A.6.

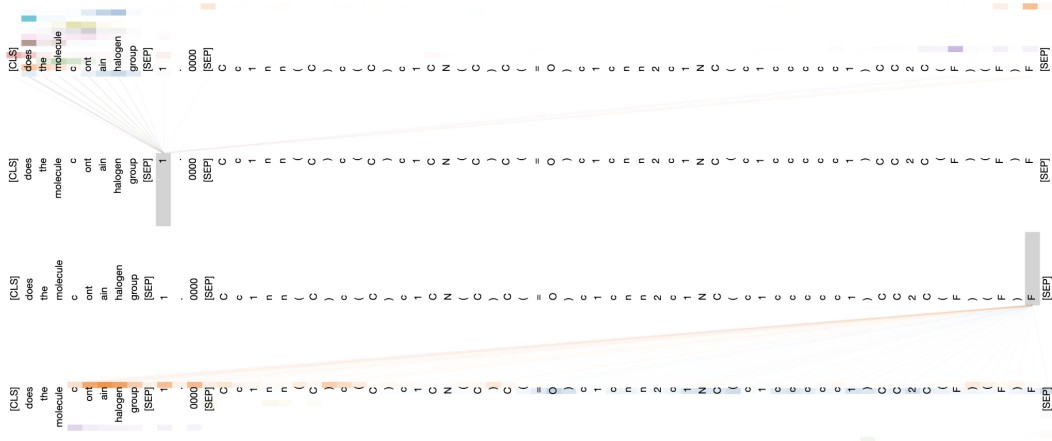

Figure 5: **Attention heads in the second to last layer exhibit the ability to correlate the task to prediction and corresponding chemical element.** Top, the source token for correct prediction is attended by the task description and all Fluorine (F) atoms. Bottom, the Fluorine atom receives attention from value tokens as well as the phrase "contains halogen group." Illustration created using BertViz (Vig, 2019).

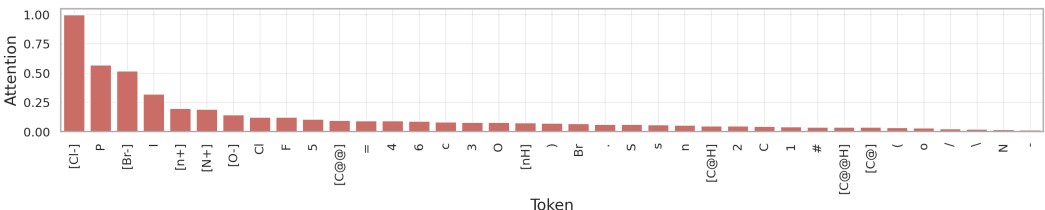

Figure 6: **Average attention per SMILES token across all attention heads for the second-to-last layer for molecules containing a halogen group.** The task description is fixed as shown in A.6 and the experiment contains 5000 molecules that in turn contain the halogen group.

### A.7 PER FUNCTIONAL GROUP EMBEDDINGS

Here we show a full embedding breakdown per functional group. The molecules are from the test set that has been scaffold split against the training set. As shown in the Fig. A.7 CILTs embeddings cluster for each of the groups (except thiol) into clusters based on the functional group.

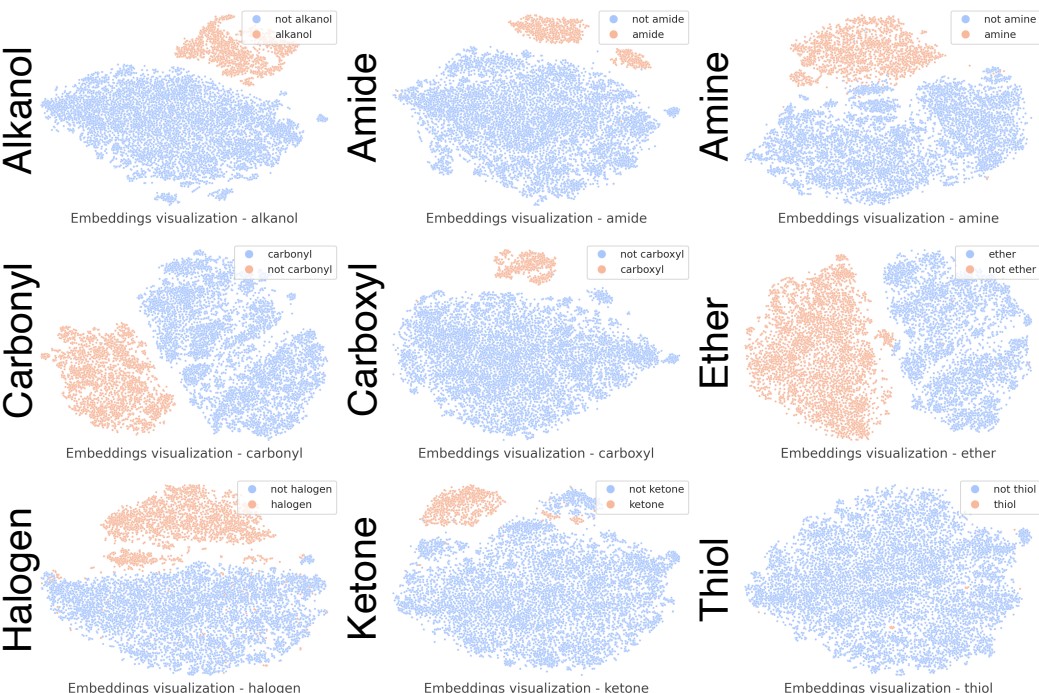

Figure 7: **Functional group embeddings breakdown.** The task description is fixed for each of the functional groups. The model is in prediction mode, where the value of the functional group is masked and the molecule is shown in full. Molecules are from the test set that is scaffold split against the train set.

## A.8 CILT Scaling Results Breakdown

Here we expand on the section 6.4 results for all of the version of CILT Models we have trained that are not shown in the main text. We conduct the same linear probe experiments as in the previously mentioned section and give a breakdown based on the number of molecules in the pre-training dataset.

Table 4: **Linear probe breakdown for different versions of CILT** Logistic regression and linear regression trained on embeddings over 4-fold cross-validation scaffold split. For classification we report %AUCROC ($\uparrow$) and for regression MAE ($\downarrow$).

| | Classification (%AUCROC $\uparrow$) | | | | | | | | |
|---|---|---|---|---|---|---|---|---|---|
| Model | BACE | BBBP | ClinTox | HIV | SIDER | Tox21 | ToxCast | MUV | Avg. |
| 75k | $78.9 \pm 4.0$ | $94.3 \pm 1.3$ | $98.1 \pm 1.6$ | $74.9 \pm 2.0$ | $57.3 \pm 6.4$ | $72.5 \pm 6.3$ | $68.4 \pm 11.0$ | $73.8 \pm 13.0$ | 77.3 |
| 125k | $79.0 \pm 3.3$ | $94.4 \pm 0.8$ | $98.3 \pm 0.1$ | $75.3 \pm 1.1$ | $58.0 \pm 6.7$ | $72.7 \pm 6.4$ | $68.1 \pm 11.2$ | $72.5 \pm 13.4$ | 77.3 |
| 250k | $81.3 \pm 2.5$ | $94.5 \pm 1.3$ | $98.3 \pm 0.1$ | $75.6 \pm 0.7$ | $58.5 \pm 6.8$ | $72.5 \pm 6.0$ | $68.0 \pm 11.2$ | $75.2 \pm 12.3$ | 78.0 |
| 375k | $78.9 \pm 3.8$ | $94.4 \pm 0.7$ | $98.7 \pm 1.1$ | $75.7 \pm 1.0$ | $58.8 \pm 5.8$ | $72.3 \pm 6.0$ | $68.6 \pm 11.0$ | $74.2 \pm 11.6$ | 77.7 |
| 500k | $80.4 \pm 1.2$ | $92.5 \pm 1.2$ | $97.7 \pm 1.5$ | $73.9 \pm 1.5$ | $55.2 \pm 6.3$ | $66.3 \pm 6.9$ | $64.4 \pm 10.3$ | $71.9 \pm 13.7$ | 75.3 |

| | Regression (MAE $\downarrow$) | | | | | | | | |
|---|---|---|---|---|---|---|---|---|---|
| Model | Lipo | ESOL | FreeSolv | CAM | PBE0 | $En - \pi*$ | $E\pi - \pi*$ | $Zn - \pi*$ | Rank |
| 75k | $0.81 \pm 0.02$ | $0.91 \pm 0.30$ | $0.90 \pm 0.18$ | $42.0 \pm 12.6$ | $66.5 \pm 7.4$ | $27.6 \pm 11.9$ | $51.3 \pm 8.0$ | $14.0 \pm 5.2$ | 3 |
| 125k | $0.81 \pm 0.02$ | $0.91 \pm 0.30$ | $0.90 \pm 0.18$ | $38.7 \pm 13.9$ | $67.7 \pm 7.9$ | $27.5 \pm 12.0$ | $51.3 \pm 7.3$ | $14.0 \pm 5.2$ | 3 |
| 250k | $0.81 \pm 0.02$ | $0.90 \pm 0.18$ | $0.91 \pm 0.30$ | $49.1 \pm 19.1$ | $65.8 \pm 7.0$ | $27.5 \pm 12.0$ | $51.3 \pm 7.3$ | $13.9 \pm 5.2$ | 4 |
| 375k | $0.81 \pm 0.02$ | $0.90 \pm 0.18$ | $0.91 \pm 0.30$ | $43.6 \pm 11.9$ | $66.4 \pm 6.8$ | $27.6 \pm 12.0$ | $51.3 \pm 7.3$ | $13.9 \pm 5.2$ | 3 |
| 500k | $0.80 \pm 0.02$ | $0.88 \pm 0.18$ | $0.91 \pm 0.30$ | $46.9 \pm 15.5$ | $58.5 \pm 7.6$ | $27.5 \pm 12.0$ | $51.3 \pm 7.3$ | $13.9 \pm 5.2$ | 2.5 |

## A.9 USE OF LLMS

Large language models were employed as assistive tools for tasks including text rewriting, spellchecking, minor stylistic improvements, and the writing of this statement. All content was reviewed and verified by the authors, who take full responsibility for the final manuscript. LLMs did not contribute to research ideation or substantive writing decisions.

