# OpenReview forum: "Chemical Priors at Scale: Efficient Foundation Models without Big Corpora"
_ICLR.cc/2026/Conference — ICLR 2026 Conference Withdrawn Submission_

### Official Review · Reviewer_qUSM · 2025-10-29

**Soundness:** 3
**Presentation:** 2
**Contribution:** 2
**Rating:** 4
**Confidence:** 4

**Summary:**

This paper describes a synthetic data augmentation strategy for pre-training transformer models on chemically-relevant information automatically extracted from molecular structure. Instead of learning from billions of molecules like MolFormer or tens of millions like ChemBERTa, they train a 150M parameter model (CILT) on ~500K molecules using hundreds of programmatically-generated chemical tasks (functional groups, substructure counts, molecular properties) expressed as natural language prompts.

The training uses dual masking: predict masked SMILES tokens conditioned on task descriptions and property values, and vice versa, predict property values conditioned on SMILES and task descriptions. The approach can be viewed as pushing property prediction post-training into the pre-training phase itself, combining what would traditionally be separate pre-training (masked language modeling on SMILES) and fine-tuning (property prediction) stages into a single joint objective.

Main claims: (1) competitive performance on MoleculeNet using 2-3 orders of magnitude fewer molecules, (2) learned representations cluster by functional groups without explicit supervision, (3) zero-shot transfer to new tasks correlates with semantic similarity of task descriptions, (4) few-shot adaptation to novel chemical tasks with <100 examples. Two theorems provide formal justification for why semantic similarity predicts transfer and why motif supervision reduces sample complexity.

**Strengths:**

**Clear introduction and related work coverage**: The paper provides a well-structured introduction to the problem and covers relevant prior work, though some recent works are missing (see weaknesses).

**Correct treatment of MoleculeNet benchmarks**: Due to the difficulty in getting autoregressive models to produce scalar-valued outputs, many recent works using pre-trained LLMs incorrectly coerce the MoleculeNet regression tasks to classification tasks, or else omit evaluations on the regression tasks entirely. The authors should be commended for correctly respecting the MoleculeNet task structure and reporting direct comparisons against relevant related models (MolCLR, ChemBERTa, MolFormer, Grover, MolBERT).

**Smart problem formulation**: The dual masking objective is elegant and the natural language task conditioning makes the framework extensible without architectural changes. This is a real practical advantage over fixed task heads.

**Solid data efficiency argument**: Figure 4 makes a compelling case that CILT achieves better parameter-performance tradeoffs than much larger models trained on orders of magnitude more data. Being competitive with models using 0.5M vs 1.1B molecules is impressive.

**Theory adds depth**: Theorems 1 and 2 provide formal justification for why semantic similarity predicts transfer (task geometry via Wasserstein bounds) and why motif supervision helps (sparse regression analysis). While the techniques are standard, the application to this setting and the connection to empirical observations strengthen the paper.

**Chemical interpretability is compelling**: The t-SNE plots (Figure 3) showing functional group clustering are visually convincing and provide clear evidence that the model learns chemically meaningful representations. The attention analysis (Appendix A.6) showing the model attending to relevant atoms (e.g., fluorine for halogen detection) is nice validation. It would be interesting to see a before/after comparison of these embeddings with standard SMILES pretraining vs CILT pretraining.

**Thorough experimental design**: The zero-shot and few-shot experiments (Figure 2) are well-designed. Testing 15 methylation variants and showing correlation between cosine similarity and adaptation speed validates the theoretical claims about task transfer.

**Weaknesses:**

**Simplistic method**: The training objective boils down to pre-training on a joint loss with a SMILES term and a property prediction term (equations 1-2). This can be seen as simply combining the pre-training and post-training stages of existing works like ChemBERTa, which pre-train on SMILES as a language modeling task and then fine-tune on property prediction. Here, the post-training on property prediction has been pushed into pre-training. While this may be effective, the technical contribution is incremental.

**Limited novelty**: The approach is very similar to other recent works which pre-train transformers on multiple auxiliary tasks extracted from molecular structure. MolX (https://arxiv.org/pdf/2406.06777v9), InstructMol (https://arxiv.org/pdf/2311.16208), and KnowMol (https://arxiv.org/abs/2510.19484v1) all use similar strategies of incorporating chemical task supervision during pre-training. The paper should more clearly articulate what distinguishes CILT from these approaches.

**Questionable core argument**: The authors argue that a weakness of existing approaches is that they treat motif recognition as an "emergent capability rather than an explicit objective" (line 54). This argument goes against the grain of prevailing wisdom in deep learning, which is that learned features are often more scalable and performant than explicitly-engineered features. The intro doesn't adequately justify why chemistry should be different.

**Incomplete data efficiency analysis**: The data efficiency argument focuses on using fewer molecules (0.5M vs 1B). However, CILT presumably generates a lot of additional text tokens from the task descriptions and property values. A more fair comparison would report the total number of tokens used in pre-training across all methods, not just molecule counts.

**Confounded data efficiency comparisons**: Figure 4 makes a compelling visual argument about data efficiency, but the comparisons are potentially confounded by multiple factors beyond just number of pretraining molecules. Different baselines use different architectures (linear attention in MolFormer, different encoder sizes), different optimization procedures, different pretraining datasets (not just size but composition and quality), and were trained at different points in time with different best practices. Without a controlled experiment holding these factors constant and varying only the pretraining approach (CILT vs standard MLM) and dataset size, it's hard to definitively attribute the gains to the task conditioning strategy rather than other methodological choices.

**Missing recent LLM baselines**: The paper only compares to older LLM-based models (MolFormer 2022, ChemBERTa 2020). More recent approaches like Text+Chem T5, MoleculeSTM, and CLAMP that also do text-molecule joint training are mentioned in related work but not included in benchmark tables. This makes it hard to assess true state-of-the-art performance.

**Missing non-LLM baselines**: There exist many strong non-LLM baselines for MoleculeNet (e.g., D-MPNN and other graph neural network approaches) which ought to be reported for completeness. Graph-based methods often outperform sequence-based methods on molecular property prediction.

**Weak regression performance**: Looking closely at Table 1, CILT isn't actually winning on regression tasks (ranks 3.8 overall vs ChemBERTa at 1.8). The "competitive" framing in the abstract is generous. The paper emphasizes being best on classification but downplays being worse on regression.

**Large variance on key results**: Standard deviations are large on some tasks (MUV, ToxCast) which makes it hard to tell if differences are meaningful. More rigorous statistical testing would strengthen the claims.

**Theoretical results are incremental**: Theorem 1 applies standard Wasserstein contraction bounds to task embeddings. Theorem 2 is standard LASSO analysis from sparse regression literature. Neither is particularly novel from a theory perspective. The assumptions (Lipschitz losses, motif-aligned representations, restricted eigenvalue condition) are strong and not validated empirically.

**Task generation lacks detail**: The paper says they generate "hundreds of programmatically-derived chemical tasks" but doesn't give enough detail on exactly what these are, how many of each type, or how they balance different categories. Are all 300+ properties used per molecule? How are templates chosen? This is important for reproducibility.

**Missing ablations**: What happens if you just use more SMILES data with standard MLM? The ablation in Table 2 trains a SMILES-only model on the same 500K molecules, but doesn't test if scaling up data for SMILES-only would close the gap. Also no ablation on the dual masking what if you only did one direction?

**Zero-shot results not that impressive**: Figure 2A shows zero-shot AUCROC around 0.5-0.7 for most similarity bins. That's only marginally better than random for binary classification. The few-shot results are better but still need 50-100 examples to converge, which isn't remarkably few.

**Venue fit**: The simplistic nature of the approach and focus on a specific domain application raises questions about whether this is the right venue. A fair case could be made that this work is of most interest to computational chemistry practitioners and might be better suited to an ACS/JCIM-type venue than ICLR, which typically values novel methodological contributions to machine learning.

**Questions:**

1. Can you provide comparisons to recent text-molecule joint training methods like MolT5, MoleculeSTM, and CLAMP? These seem like the most relevant baselines but are missing from benchmark tables.

2. Can you report strong non-LLM baselines like D-MPNN for completeness? How does CILT compare to graph-based methods which often excel at molecular property prediction?

3. What is the total token count comparison between CILT and baseline methods? The claim of data efficiency focuses on molecule counts, but CILT generates substantial additional text tokens from task descriptions and property values. A fair comparison should account for total tokens.

4. Can you run a controlled experiment to validate the data efficiency claims? Specifically, train both CILT and standard SMILES-only pretraining (same architecture, same optimization, same compute budget) on increasing dataset sizes (e.g., 100K, 500K, 1M, 5M molecules) and plot the downstream performance curves. This would isolate the effect of task conditioning from other confounding factors like architecture choices, dataset composition, and training procedures that differ across the baselines in Figure 4.

5. How sensitive are results to the specific set of pretraining tasks? What happens if you use only functional groups vs only global properties? Is there a minimum diversity of task types needed? Can you provide ablations breaking down contribution by task category?

6. Can you provide a before/after comparison showing embedding structure with standard SMILES pretraining vs CILT pretraining? This would help validate that the improved clustering is due to your method rather than general transformer learning.

---

> ### Author Response · Authors · 2025-11-26
>
> We thank the reviewer for taking the time to read our manuscript and providing thorough feedback. We have reworked the introduction and added a background section to better highlight the chemical and theoretical foundations of our approach. CILT employs important chemical motifs as a weak supervision prior, enabling higher levels of interpretability and adaptability. We showcase interpretability by adding comparisons of embedding clusterings in Figure 2 and Appendix A7, where CILT is the only model capable of clustering embeddings in a chemically meaningful way. We showcase the higher level of adaptability by incorporating the MLM-trained models in Figure 3, where CILT is the only model capable of performing better than random. In addition to this, we have added additional benchmark models to Tables 1 & 2, as well as the scaled version of CILT with a full breakdown in Appendix A8. Below, we respond in more detail to reviewers' points that we incorporated into our manuscript.
>
> **Our revised manuscript has been uploaded to open review.**
>
> ---
>
> **Simplistic method: The training objective boils down to pre-training on a joint loss with a SMILES term and a property prediction term (equations 1-2). This can be seen as simply combining the pre-training and post-training stages of existing works like ChemBERTa, which pre-train on SMILES as a language modeling task and then fine-tune on property prediction. Here, the post-training on property prediction has been pushed into pre-training. While this may be effective, the technical contribution is incremental.**
>
> We propose a method that performs weakly supervised pretraining on tasks inspired by group contribution methods.
> This is in scale and diversity, different from combining pre- and post-training stages and leads to different embeddings as shown in the new Figure 3
> To highlight this, we revised the introduction and added a new background section explaining group contribution methods.
> In addition, our weakly supervised training techniques also allow us to use the model in a task-conditioned way, which has been impossible with other chemical property prediction models. This ability leads to zero- and few-show adaptability that greatly outperforms other models.
>
> **Limited novelty: The approach is very similar to other recent works which pre-train transformers on multiple auxiliary tasks extracted from molecular structure. MolX (https://arxiv.org/pdf/2406.06777v9), InstructMol (https://arxiv.org/pdf/2311.16208), and KnowMol (https://arxiv.org/abs/2510.19484v1) all use similar strategies of incorporating chemical task supervision during pre-training. The paper should more clearly articulate what distinguishes CILT from these approaches.**
>
> We employ chemically important motifs as a weak supervision prior. Our approach leads to a chemically informed model, which is shown in the updated Figure 2, where CILT is the only model capable of separating molecules in a chemically meaningful way. Additionally, we show in the updated Figure 3 that other models can not perform better than random, even after 100 examples, while our model can reach 100% accuracy. In addition to this, we have reworked the introduction and added a background section.
>
> **Questionable core argument: The authors argue that a weakness of existing approaches is that they treat motif recognition as an "emergent capability rather than an explicit objective" (line 54). This argument goes against the grain of prevailing wisdom in deep learning, which is that learned features are often more scalable and performant than explicitly-engineered features. The intro doesn't adequately justify why chemistry should be different.**
>
> We have removed the following statement and revised the introduction section. We argue that we can weakly supervise models to understand chemical motifs with orders of magnitude fewer molecules and increased interpretability. Other models try to leverage scale to capture chemically relevant features (“emergent capability“) and fail at it (see Figures 2 & 3) while CILT learns directly on them (explicit objective), while maintaining flexibility, and shows equivalent if not better performance.
> This means that our weakly supervised pretraining technique can be thought of as providing CILT with a soft inductive bias. Conventional fingerprints and descriptors are hard inductive bias that limit the expressivity of the model. CILT provides an inductive bias toward expressing chemical properties in terms of group contribution (which is a meaningful decomposition as described in the new background section) while maintaining full expressivity.

---

> ### Author Response · Authors · 2025-11-26
>
> **Missing recent LLM baselines: The paper only compares to older LLM-based models (MolFormer 2022, ChemBERTa 2020). More recent approaches like Text+Chem T5, MoleculeSTM, and CLAMP that also do text-molecule joint training are mentioned in related work but not included in benchmark tables. This makes it hard to assess true state-of-the-art performance.**
>
> We have updated Table 1 in the revised manuscript with the additional LLM-based models.
>
> **Missing non-LLM baselines: There exist many strong non-LLM baselines for MoleculeNet (e.g., D-MPNN and other graph neural network approaches) which ought to be reported for completeness. Graph-based methods often outperform sequence-based methods on molecular property prediction.**
>
> We are not able to compare D-MPNN, since it is not a pre-trained model; it needs to be trained towards a specific task, which makes the logprobs an improper way of comparing it against other models (please refer to the GitHub page of the original work for clarification https://github.com/chemprop/chemprop/issues/933).
>
> **Task generation lacks detail: The paper says they generate "hundreds of programmatically-derived chemical tasks" but doesn't give enough detail on exactly what these are, how many of each type, or how they balance different categories. Are all 300+ properties used per molecule? How are templates chosen? This is important for reproducibility.**
>
> The task generation, as stated in the paper, was done utilizing the ChemCaption package, which we have added to open source as one of the contributions of our work. All the molecules use all the features. We utilize all of the featurizers from ChemCaption. All of the supporting data and generation scripts will be released with the non-annomized version of the manuscript.
>
> **Zero-shot results not that impressive: Figure 2A shows zero-shot AUCROC around 0.5-0.7 for most similarity bins. That's only marginally better than random for binary classification. The few-shot results are better but still need 50-100 examples to converge, which isn't remarkably few.**
>
> We have updated the zero-shot results with the MLM pre-trained version of CILT and two other non-motif aligned models (MolFormer & MolT5), showcasing that they are not capable of performing better than random while utilizing 100 or fewer data points for fine-tuning.
>
> **Can you provide a before/after comparison showing embedding structure with standard SMILES pretraining vs CILT pretraining? This would help validate that the improved clustering is due to your method rather than general transformer learning.**
>
> We have updated Figure 2 with the SMILES-only pre-training of CILT and added the embeddings visualizations from MolT5 and MolFormer. CILT is the only model capable of separating molecules in a chemically meaningful way.

---

### Official Review · Reviewer_95et · 2025-10-31

**Soundness:** 2
**Presentation:** 1
**Contribution:** 2
**Rating:** 2
**Confidence:** 3

**Summary:**

The paper presents CILT, a 150M-parameter ModernBERT trained with task-conditioned self-supervision: hundreds of programmatically generated chemistry tasks are phrased in natural language and used to (a) predict masked SMILES from a task description and property value and (b) predict property values from a SMILES and task description. The goal is to unify generation and property prediction while achieving better data-efficiency than prior molecular foundation models, with minimal architectural changes.

**Strengths:**

* Intuitive idea: inject known chemistry priors via programmatically-derived chemical tasks with minimal architectural changes.
* Encouraging signs of data-efficiency on several classification benchmarks.

**Weaknesses:**

* My main concern is on presentation. For example, Theorem 1 is confusing because key notions are not defined in the main text. What precisely is the “risk” in Theorem 1? Are “source vs. target tasks” meant to be pretraining vs. inference tasks? Does alpha denote per-task weights, and how is it quantified? What is h? While the detailed proofs can remain in the appendix, the main text should still deliver the full insights.
* In line 218, the claim that transfer improves with higher cosine similarity between task embeddings might be impractical if the weighted source-risks term dominate the bound’s RHS, which will yield negligible gains even for similar embeddings.
* In Theorem 2, “motif-aligned pretrained representation” (line 232) is undefined. When does k<< p (k motifs vs. p features) actually hold? Because for some molecules (e.g. small molecules), many or all features can drive a property (k≈p). The stated reduction from O(p/e^2) to O(k log p / e^2) then does not hold on these type of molecules.
* There is no comparison to baselines pretrained on less data. Current results show CILT is competitive when trained on less data, but do not establish that existing baselines' pretraining schemes would not remain as competitive under similar data budgets.
* In Line 361, showing that CILT can achieve competitive performance with <100-sample fine-tuning and zero-shot performance does not by itself validate “motif alignment leads to more data-efficient learning.” A more direct test would compare motif-aligned vs. non-motif-aligned variants of CILT, and also against models that take motifs directly as inputs (rather than SMILES). If Theorem 2 holds, such motif-native models should also be data-efficient.

**Questions:**

Please refer to weaknesses.

---

> ### Author Response · Authors · 2025-11-26
>
> We thank the reviewer for taking the time to read our manuscript and providing constructive feedback. In the revised manuscript, we have updated both of the theorems and added context to them with a new background section.
> The first theorem now clearly defines core notation in the main text, and the second theorem points to the group contribution method as the source for the k<<p assumption. In addition, we have updated Figures 2 & 3 to include comparison models showing clearly the advantages of our work. We have added a background section to connect the motivation and theory more clearly.
>
> **Our revised manuscript has been uploaded to open review.**
>
> ---
>
> **My main concern is presentation. For example, Theorem 1 is confusing because key notions are not defined in the main text. What precisely is the “risk” in Theorem 1? Are “source vs. target tasks” meant to be pretraining vs. inference tasks? Does alpha denote per-task weights, and how is it quantified? What is h? While the detailed proofs can remain in the appendix, the main text should still deliver the full insights.**
>
> The notations are fixed in the revised manuscript. All the terms in Theorem 1 are now clearly explained in the main text.
>
> **In line 218, the claim that transfer improves with higher cosine similarity between task embeddings might be impractical if the weighted source-risks term dominate the bound’s RHS, which will yield negligible gains even for similar embeddings.**
>
> If the RHS is dominated by the source risk term, then the model already performs badly in the source domain. We have updated the claim so that the higher cosine does not always imply better transferability.
>
> **In Theorem 2, “motif-aligned pretrained representation” (line 232) is undefined. When does k<< p (k motifs vs. p features) actually hold? Because for some molecules (e.g. small molecules), many or all features can drive a property (k≈p). The stated reduction from O(p/e^2) to O(k log p / e^2) then does not hold on these types of molecules.**
>
> This assumption comes from the group contribution method. The revised manuscript makes this clear, and the addition of a background section describes this method with long tradition in chemistry further.
>
> **In Line 361, showing that CILT can achieve competitive performance with <100-sample fine-tuning and zero-shot performance does not by itself validate “motif alignment leads to more data-efficient learning.” A more direct test would compare motif-aligned vs. non-motif-aligned variants of CILT, and also against models that take motifs directly as inputs (rather than SMILES). If Theorem 2 holds, such motif-native models should also be data-efficient.**
>
> We have updated Figure 3 with the SMILES-only version of CILT and other non-motif aligned models. The figure now shows clearly that non-motif aligned models are not capable of achieving better than random performance while using 100 or fewer fine-tuning samples.

---

### Official Review · Reviewer_2jUH · 2025-11-01

**Soundness:** 2
**Presentation:** 3
**Contribution:** 2
**Rating:** 4
**Confidence:** 3

**Summary:**

This paper presents CILT (Chemically Informed Language Transformer), a 150M-parameter transformer that aims to achieve competitive molecular property prediction performance using significantly fewer pretraining molecules than existing methods. The approach replaces standard masked language modeling with task-conditioned pretraining on hundreds of programmatically-derived chemical tasks (functional groups, substructure counts, molecular properties) expressed as natural language descriptions. During pretraining, the model alternates between predicting masked SMILES tokens conditioned on task descriptions and predicting property values conditioned on molecules. The authors provide theoretical analysis suggesting that semantic similarity between task descriptions controls transfer performance and that motif-based pretraining reduces sample complexity. Experiments on MoleculeNet benchmarks demonstrate competitive performance with state-of-the-art models while using 2-3 orders of magnitude fewer molecules.

**Strengths:**

- The method demonstrates competitive performance with fewer pretraining molecules, which is a valuable property for data-constrained settings.
- The evaluation suite is quite comprehensive, covering classification and regression tasks across MoleculeNet and photoswitch datasets. The analysis of learned representations (methylations experiment, embedding clustering by functional groups, attention patterns) provides valuable insights into model behavior.
- Improving data efficiency for molecular foundation models is a valuable research direction, particularly for practical applications where obtaining labeled data is expensive.
- The paper is generally well-written and easy to follow, with clear figures and informative visualizations.

**Weaknesses:**

- The core contribution is essentially augmenting SMILES sequences with task descriptions and property values, then applying bidirectional masking. This is a relatively incremental modification to existing molecular language modeling approaches.
    - Further, given that standard MLM (potentially with some specialized masking around task descriptions) on sequences containing [task description] [property value] [SMILES] should implicitly learn the correlations between all components, it is unclear why explicitly splitting into two separate masking objectives is necessary or beneficial. The paper does not provide ablations or justification for this design choice versus standard MLM on the full concatenated sequence.
- Several essential training details are absent:
    - What constitutes an "epoch"? Is it a single pass over all task-molecule pairs (\~150M training steps) or random task sampling per molecule per epoch (\~500k steps)? This dramatically affects total training computation.
    - How are epochs defined for the SMILES-only baseline? Without this, the ablation comparison is not interpretable.
    - Why not report standardized compute metrics (FLOPs, GPU hours) to substantiate claims about computational efficiency?
- The theoretical section (Section 3.3) appears somewhat disconnected from the main contribution:
    - Theorem 2 on motif sample complexity essentially states that if we assume molecular properties depend on sparse motifs, then training on motifs improves sample efficiency. The analysis is straightforward given the sparsity assumption.
        - More critically, if this assumption holds, would it not be more effective to explicitly provide motifs as input features rather than relying on implicit learning through pretraining? This seems like a missed opportunity to design a more effective method.
    - While the theorems provide formal justification for intuitive priors, they do not offer particularly actionable insights for method design. The value of this work lies primarily in the empirical contributions rather than theoretical novelty. Consider moving this section to an appendix.
- The paper presents the loss function as a weighted combination of SMILES and property prediction losses $\mathcal{L}(\theta) = \mathcal{L}_{\text{smiles}}(\theta) + \lambda \mathcal{L}_{\text{value}}(\theta)$ (Section 3.2), implying joint optimization. However, the actual training alternates between these objectives every 20 batch steps (Section 4.2). If the loss is not optimized jointly, the joint formulation in Section 3.2 is misleading and should be corrected.
- The paper acknowledges that scaling experiments are "left for future work," but this is arguably the most important validation for a foundation model approach. Without evidence that the approach scales favorably, the practical impact remains uncertain.

**Questions:**

1. Can you clarify the exact training procedure? Specifically:
    - Is the loss in Equation (3) actually optimized jointly, or is the alternating schedule the true training objective?
    - What constitutes an epoch in your experiments?
    - What is the total number of training steps for CILT versus baselines?
    - What is the exact number of pretraining molecules for each baseline?
    - Are the pretraining data distributions comparable across methods?
2. How do you measure closeness between pretraining data and downstream tasks? For molecular property prediction, this could involve comparing chemical composition distributions, motif distributions, etc.
    - This matters for assessing whether apparent efficiency gains simply reflect better data-task alignment.
3. The maximum sequence length of 1024 seems quite limited given the latest transformer models that handle much longer sequences. Further, given your specific tokenization scheme, I suspect that this could also be a bottleneck for larger molecules. Can you justify this choice and discuss its implications?
4. Can you provide an ablation comparing your alternating masking approach against standard MLM on the concatenated sequence [task description] [property value] [SMILES]? This would help justify the design choice.
5. Can you report standardized compute metrics (FLOPs or GPU hours) for all models to substantiate the efficiency claims?
6. (More open-ended/curiousity, not a critique) Have you considered or experimented with LLM-based approaches (fine-tuning pretrained language models)? Given that you're using natural language task descriptions, it seems like a natural alternative to fine-tune a pretrained LLM that has already seen extensive chemistry-related text during pretraining. While the bidirectional nature of CILT enables flexible masking that would be challenging with autoregressive LLMs, this tradeoff deserves consideration. Works like "Fine-Tuned Language Models Generate Stable Inorganic Materials as Text" have demonstrated promising results in this direction (for the materials domain).

---

> ### Author Response · Authors · 2025-11-26
>
> We thank the reviewer for taking the time to read our manuscript and providing constructive and helpful feedback. We have reworked the manuscript to incorporate the reviewers' feedback. We have reworked the theory section and added a background section covering the group contribution theory and molecular fingerprints to better showcase the connection to our approach. Additionally, we have added the comparison in Figures 2 & 3, where we showcase that our model adapts with fewer examples and is capable of clustering embeddings in a chemically meaningful way. We have added the scaled versions of CITL in Tables 1 and Figure 4, with the full breakdown in Appendix A7. Below, we respond in more detail to the points brought up by the reviewer.
>
> **Our revised manuscript has been uploaded to open review**
>
> ---
>
> **The core contribution is essentially augmenting SMILES sequences with task descriptions and property values, then applying bidirectional masking.**
>
> The core contribution of our work is weak supervision on chemically important motifs, which yields an interpretable model requiring only a fraction of molecules for training. In the revised manuscripts, we have made this clearer by providing additional comparisons of MLM-trained models in the adaptation experiments visible in Figure 3, where our model is the only one capable of performing better than random. Additionally, we compare the embedding clustering by chemical features with other MLM models, where our model is the only one capable of separating molecules in a chemically meaningful way (Figure 2 & Appendix A7). We offer a layer of chemical interpretability that deep learning models are not capable of.
>
> **The theoretical section (Section 3.3) appears somewhat disconnected from the main contribution**
>
> Section 3.3 (now section 4.3)  is the main idea behind the contribution of our model; we have reworked the introduction and added a background section to clarify this further. For Proof 1, we have clarified the language to align better with the new introduction and added the explanation of notation to the main text. For Proof 2, we have added a background section covering group contribution method and molecular fingerprints, making the connection between chemical theory and model design choices clearer.
>
> **“More critically, if this assumption holds, would it not be more effective to explicitly provide motifs as input features rather than relying on implicit learning through pretraining”.**
>
> We do not specifically provide motifs as features, as we aim for a flexible model. To clarify this further, we have now added a background section covering molecular fingerprints, an approach that featurizes molecules into fixed feature vectors. Our work incorporates molecular fingerprints as weak supervision, leveraging the flexibility of deep learning with the interpretability of molecular fingerprints.
>
> **The paper presents the loss function as a weighted combination of SMILES and property prediction losses $\mathcal{L}(\theta) = \mathcal{L}{\text{smiles}}(\theta) + \lambda \mathcal{L}{\text{value}}(\theta)$ (Section 3.2), implying joint optimization.**
>
> The loss has been updated in the revised manuscript to reflect the alternating objective.
>
> **The paper acknowledges that scaling experiments are "left for future work," but this is arguably the most important validation for a foundation model approach. Without evidence that the approach scales favorably, the practical impact remains uncertain.**
>
> Our new results show that CILT scales favourably with the number of molecules, showing competitive performance with 75k, 12k, and 250k molecules. We have added the scaled models to Tables 1 & 2, as well as Figure 5; the full breakdown is visible in Appendix A8.
>
> **How do you measure closeness between pretraining data and downstream tasks? For molecular property prediction, this could involve comparing chemical composition distributions, motif distributions, etc.**
>
> For the Zero-shot transfer, we measure the cosine similarity between the property task descriptions, and for the few-shot transfer, we conduct methylations, replacing one H with CH3.
> For the majority of downstream tasks, it is hard to estimate how and if a specific motif contributes to the property, especially, for example, with activity cliffs (see https://pubs.acs.org/doi/10.1021/acs.jcim.2c01073), where small changes affect the prediction greatly. Therefore, we have focused on well-known and studied functional groups and substructures, giving our model a good baseline of chemical knowledge.

---

> ### Author Response · Authors · 2025-11-26
>
> **The maximum sequence length of 1024 seems quite limited given the latest transformer models that handle much longer sequences. Further, given your specific tokenization scheme, I suspect that this could also be a bottleneck for larger molecules. Can you justify this choice and discuss its implications?**
>
> In the revised manuscript we highlight that no sequence in our experiments exceeds the length of 1024. The larger inputs — which we consider unlikely to occur for the applications we target — would require re-training or context extension.
>
> **Can you provide an ablation comparing your alternating masking approach against standard MLM on the concatenated sequence [task description] [property value] [SMILES]? This would help justify the design choice.**
>
> Born, J., & Manica, M. (ref. line 529 in our manuscript) performed the ablation which has been the foundation for our design choice.
>
> **Can you report standardized compute metrics (FLOPs or GPU hours) for all models to substantiate the efficiency claims?**
>
> We have added the computer metrics for CILT in the Appendix A5.

---

### Comment · Area_Chair_Uv9G · 2025-11-27
**Request for Timely Response to Authors’ Rebuttal and Discussion**

Dear Reviewers,

I hope you are doing well. The authors have now submitted their rebuttal for the paper under your review. At this stage, your timely response is essential for ensuring a smooth discussion phase.

Could you please review the rebuttal at your earliest convenience and share your updated thoughts? If there are points that require further discussion among the reviewers, please feel free to initiate or join the conversation on the discussion thread.

Your prompt input will greatly help us maintain the review timeline. Thank you very much for your efforts and valuable contributions.

Best regards,

AC

---

### Author Response · Authors · 2025-11-29

We thank all reviewers and the AC for the time and effort dedicated to evaluating our manuscript. We have made major revisions to the manuscript in line with the reviewers' comments. Below we summarise all of the changes.

Important Concerns Raised by the Reviewers
===

**Clarification of core contributions**

We clarified that the central contribution of CILT is the use of weak supervision on chemically meaningful motifs during pretraining, which leads to interpretable, motif-aligned representations and significantly improved data efficiency. We revised the introduction, added a dedicated background section covering the group contribution method and molecular fingerprints. With these changes, we highlight the core contribution of our work and distinguish it from previous approaches.

**Substantial additions to experiments**

We have extended results in Figures 2 and 3 to compare CILT against MLM-trained models and a SMILES-only version of CILT. These results highlight two key properties of our method:
- *Embedding interpretability*: Only CILT forms chemically meaningful clusters (Figure 2 and Appendix A7).
- *Adaptability*: CILT is the only model that consistently outperforms a random baseline when fine-tuned with ≤100 examples (Figure 3).

Additionally, we have added scaled versions of CILT (75k, 125k, 250k molecules…) to Tables 1–2 and Figures 4, with a full breakdown in Appendix A8. These results show favorable scaling trends.

**Expanded theoretical clarification**

The theory section (previously Section 3.3, now Section 4.3) has been restructured to improve readability. We now define all notation in the main text, explain the risk terms, and provide clearer connections between theory and model design. The new background section covers the group contribution method, which supports the k ≪ p assumption and motivates motif-aligned representations.

**Clarification of design choices**

Born, J., & Manica, M. (ref. line 529 in our manuscript) performed the ablation, which has been the foundation for our design choice behind the alternating training. We have further clarified the pre-training procedure and incorporated a full breakdown of hyperparameters in Appendix A5.

**Additions to baselines and comparisons**

We have updated Table 1 to include MoleculeSTM and MolT5, and we now incorporate 7 comparison models. We were not able to incorporate D-MPNN since it is not a pre-trained model, and therefore, we are not able to evaluate logprobs (please refer to the GitHub page of the original work for clarification https://github.com/chemprop/chemprop/issues/933).

**Expanded detail on task generation and reproducibility**

We clarify that tasks are programmatically derived using the ChemCaption package. All molecules are featurized with all available ChemCaption descriptors. We will release full data along with the open source package with the de-anonymized version.

**Compute reporting**
We added standardized compute metrics (GPU hours) for CILT in Appendix A5.

We sincerely appreciate the reviewers’ detailed and constructive feedback. The resulting manuscript is significantly stronger in clarity, motivation, theoretical grounding, and empirical validation.

Key Strengths Highlighted by the Reviewers
===

**Data efficiency gains**

Reviewers **2jUH**, **qUSM** and **95et** recognize the model's data efficiency gains.

**Comprehensive evaluation suite**

Reviewers **2jUH** and **qUSM** note that the evaluation suite is comprehensive, covering regression, classification and analysis of learned representations (methylations experiment, embedding clustering by functional groups, attention patterns).

---

### Note · Authors · 2026-01-08

I have read and agree with the venue's withdrawal policy on behalf of myself and my co-authors.